**Data Availability Statement:** All relevant data are within the paper and its Supporting Information files.

**Funding:** The Centre for Higher Degrees by Research (CHDR) of Taylor's University provided a

# Risk factors associated with nosocomial infections among end stage renal disease patients undergoing hemodialysis: A systematic review

**Saad Hanif Abbasi, Raja Ahsan Aftab***, **Siew Siang Chua**

Faculty of Health and Medical Sciences, School of Pharmacy, Taylor's University, Subang Jaya, Malaysia

* rajaahsan.aftab@taylors.edu.my

## Abstract

### Background

Profound healthcare challenges confront societies with an increase in prevalence of end-stage renal disease (ESRD), which is one of the leading causes of morbidity and mortality worldwide. Due to several facility and patient related factors, ESRD is significantly associated with increased morbidity and mortality attributed to infections.

### Aims and objective

The aim of this study was to assess systematically the characteristics of patients and risk factors associated with nosocomial infections among ESRD patients undergoing hemodialysis.

### Methods

A systematic literature search was performed to identify eligible studies published during the period from inception to December 2018 pertaining to risk factors associated with nosocomial infections among hemodialysis patients. The relevant studies were generated through a computerized search on five databases (PubMed, EBSCOhost, Google Scholar, Science-Direct and Scopus) using the Mesh Words: nosocomial infections, hospital acquired infections, healthcare associated infections, end stage renal disease, end stage renal failure, hemodialysis, and risk factors. The complete protocol has been registered under PROS-PERO (CRD42019124099).

### Results

Initially, 1411 articles were retrieved. Out of these, 24 were duplicates and hence were removed. Out of 1387 remaining articles, 1337 were removed based on irrelevant titles and/or abstracts. Subsequently, the full texts of 50 articles were reviewed and 41 studies were excluded at this stage due to lack of relevant information. Finally, nine articles were selected for this review. Longer hospital stay, longer duration on hemodialysis, multiple catheter

fellowship to SHA. The funder had no role in study design, data collection and analysis, decision to publish, or preparation of the manuscript.

**Competing interests:** The authors have declared that no competing interests exist.

sites, longer catheterization, age group, lower white blood cell count, history of blood transfusion, and diabetes were identified as the major risk factors for nosocomial infections among hemodialysis patients.

## Conclusion

The results of this review indicate an information gap and potential benefits of additional preventive measures to further reduce the risk of infections in hemodialysis population. Moreover, several patient-related and facility-related risk factors were consistently observed in the studies included in this review, which require optimal control measures.

## Introduction

Profound health care challenges confront societies with an increase in prevalence of end-stage renal disease (ESRD), which is one of the leading causes of morbidity and mortality throughout the world [1]. It has been estimated that the global prevalence of patients undergoing maintenance hemodialysis (HD) increased by 1.7 times from 165 patients per million population (pmp) in 1990 to 284 patients pmp in 2010 [2]. Moreover, the global prevalence of ESRD and use of renal replacement therapy (RRT) have been projected to more than double by the next decade [1, 3]. Chronic diseases, such as ESRD, limit daily activities and affect health-related quality of life (HRQoL) of patients. Cardiovascular diseases (CVD) and infections are the major causes of deaths among these patients, together accounting for up to 70% of all deaths [4].

ESRD is significantly associated with higher mortality attributed to infections, which account for almost 20% of all deaths in these patients [4]. These patients have frequent hospital visits and more extended hospital stays, which make them more vulnerable to nosocomial infections [5]. It is possible that this complication is related to alterations in the immune system in ESRD, as uremia is linked with a state of immune dysfunction characterized by immunosuppression [4]. In addition, patients with immunodeficiency who undergo RRT could be malnourished, and the resulting imbalance in bacteria, viruses, fungi, and other microorganisms in the body could also increase the risk of nosocomial infections [6]. These infections not only affect the quality of life (QOL) of ESRD patients, but also add extra economic burden [7].

Risk factors associated with nosocomial infections in ESRD patients who are on HD are still poorly understood, and the clinical epidemiology of these infections has still not been defined adequately in the previous literature. Therefore, this review was conducted to assess systematically the characteristics of patients and risk factors associated with nosocomial infections in these patients who were receiving HD.

## Methods

We systematically identified the articles related to risk factors associated with nosocomial infections among ESRD patients, which had been published in scientific literature during the period from inception to December 2018. We followed the PRISMA (Preferred Reporting Items for Systematic Reviews and Meta-Analysis) guidelines for the preparation of this review [8]. The protocol of this systematic review has been registered under PROSPERO (CRD42019124099).

## Data sources and search methods

Relevant papers published in the literature were generated through a computerized search on five databases (PubMed, EBSCOhost, Google Scholar, ScienceDirect and Scopus) using the following Mesh Words: nosocomial infections, hospital acquired infections, healthcare associated infections, end stage renal disease, end stage renal failure, HD, and risk factors. These Mesh words search was performed through title and abstract only. For PubMed search, the Boolean search of (("Cross Infection"[Mesh]) AND ("Kidney Failure, Chronic"[Mesh] OR "Renal Dialysis"[Mesh] OR "Hemodialysis, Home"[Mesh] OR "Hemodialysis Units, Hospital"[Mesh]) AND ("Risk Factors"[Mesh])) was used. The same search terms were used for EBSCOhost, Google Scholar, ScienceDirect, and Scopus databases.

## Inclusion/Exclusion criteria

Studies on risk factors associated with nosocomial infections among ESRD patients undergoing HD published during the period from inception to December 2018 were included in this review. Moreover, studies conducted on ESRD patients who were on HD with one or more episodes of nosocomial infections were included. Only quantitative studies and studies done on patients above 12 years of age were considered. Systematic literature reviews, abstracts, scientific correspondence, posters, animal studies, case reports, advertisements, thesis, and opinions were excluded. In addition, studies conducted on ESRD patients with infections other than nosocomial infections were also excluded. Qualitative studies and studies published in a language other than English were not included in this review. Similarly, studies without clearly stated outcomes or studies conducted in pregnant patients were excluded.

## Data extraction (Selection and coding)

Retrieved articles, after preliminary searches, were imported to Endnote X7 and the duplicates were removed. Two reviewers, SHA and RAA, independently evaluated the articles for eligibility through screening of the titles and abstracts. After a preliminary screening, a full-text assessment was carried out for the final selection of articles. Any disagreement regarding the eligibility of studies between the two authors was resolved through agreement and discussion in the team meetings. Finally, all authors agreed with the selection of the final studies for the review. One reviewer (SHA) rechecked selected studies for the validation of screening procedure.

A data extraction form was developed to retrieve all the relevant information from the selected articles. The items of the data extraction form were finalized after mutual agreement between two authors SHA and RAA. These items included, author's name, year and duration of the study, study design, respondents, demographics of the patients (sample size, gender, and age), type of nosocomial infections studied, type of pathogens involved, and risk factors associated with nosocomial infections.

## Risk of bias/Quality assessment

Risk of bias (quality) assessment was undertaken by two reviewers (SHA and RAA) by using Newcastle Ottawa scale (NOS) for observational studies [9]. In Newcastle Ottawa scale, stars are awarded for three categories: "Selection," "Comparability," and "Outcome," and each of these three categories was divided into subcategories. For each study, a maximum one star could be awarded for each subcategory. However, a maximum of two stars could be given for "Comparability" category [9]. The maximum number of 9 stars could be achieved for a single study, which indicates the complete the absence of any bias [9]. The Newcastle Ottawa scale

scores were later converted into Agency for Healthcare Research and Quality (AHRQ) standards (good, fair, and poor), for final quality assessment of the studies. The number of stars for each study was calculated and this indicated the quality of the study as good, fair and poor. For good quality, a study must have 3 or 4 stars, 1 or 2 stars and 2 or 3 stars in 'Selection', 'Comparability' and 'Outcome' domains respectively. Similarly, for fair quality, a study must 2 stars, 1 or 2 stars, and 2 or 3 stars in 'Selection', 'Comparability' and 'Outcome' domains respectively. Finally, for poor quality, a study must have 0 or 1 star, 0 star, or 0 or 1 star in 'Selection', 'Comparability' and 'Outcome' domains respectively [9, 10]. Decision on quality and eligibility of studies was based on consensus, and SSC acted as an adjudicator in case of any disagreement.

## Data synthesis

A systematic review was performed to make sure that all data synthesis done was sourced from the maximum possible, and complete collection of relevant literature. Furthermore, a reference check and contact with reference authors were also performed to identify relevant work and studies. Only qualitative analysis was undertaken.

## Results

Initially, 1411 studies were retrieved. Out of these, 1039 studies were retrieved from EBSCO-host, 193 from ScienceDirect, 68 from Scopus, 66 from Google scholar, and 45 from PubMed database. Twenty four (24) of these 1411 studies were duplicates, and hence they were excluded. The remaining 1387 studies were screened based on titles and abstracts, of which 1337 articles with irrelevant titles and/or abstracts were removed. Subsequently, the full texts of remaining 50 articles were reviewed, where 41 of these studies were removed as they did not contain the relevant information or failed to meet our inclusion criteria. Eventually, nine articles were selected for this review (Fig 1).

### Evaluation of study quality

The quality rating of majority of the studies in this review was fair [11–16], while three studies had a maximum score, and were rated as good quality studies [17–19]. Table 1 shows the detail quality scoring of the studies.

### Characteristics of the selected studies

The major characteristics of 9 studies included in this review are described in Table 2. Two studies were conducted in the United States (US) [11, 12], and one each in of the China [17], South Korea [13], Canada [14], Taiwan [15], Indonesia [16], Iraq [18], and India [19]. Five studies utilized cross-sectional [12, 13, 15, 16, 18] study designs, one was retrospective cohort and case control study [11], and two were longitudinal or prospective studies [14, 19]. Tang et al. 2016, adopted a mixed study design (incorporating both cross-sectional and longitudinal study designs) [17]. There were large variations in sample size of the included studies, ranging from 105 [13] to 890 participants [17]. In a case-control study conducted by Erika et al. 2000, only 29 cases and 29 controls were included [11].

Most of the studies included only elderly people undergoing HD. The mean age was utilized for the analysis of all the studies. Erika et al. 2000, conducted a case control study, whereby ESRD patients undergoing HD with and without nosocomial infections were assigned to case and control groups respectively [11]. In the study by Wang et al. 2016, the respondents were patients with nosocomial infections, and further subgroups were formed based on patients with and without HD [15].

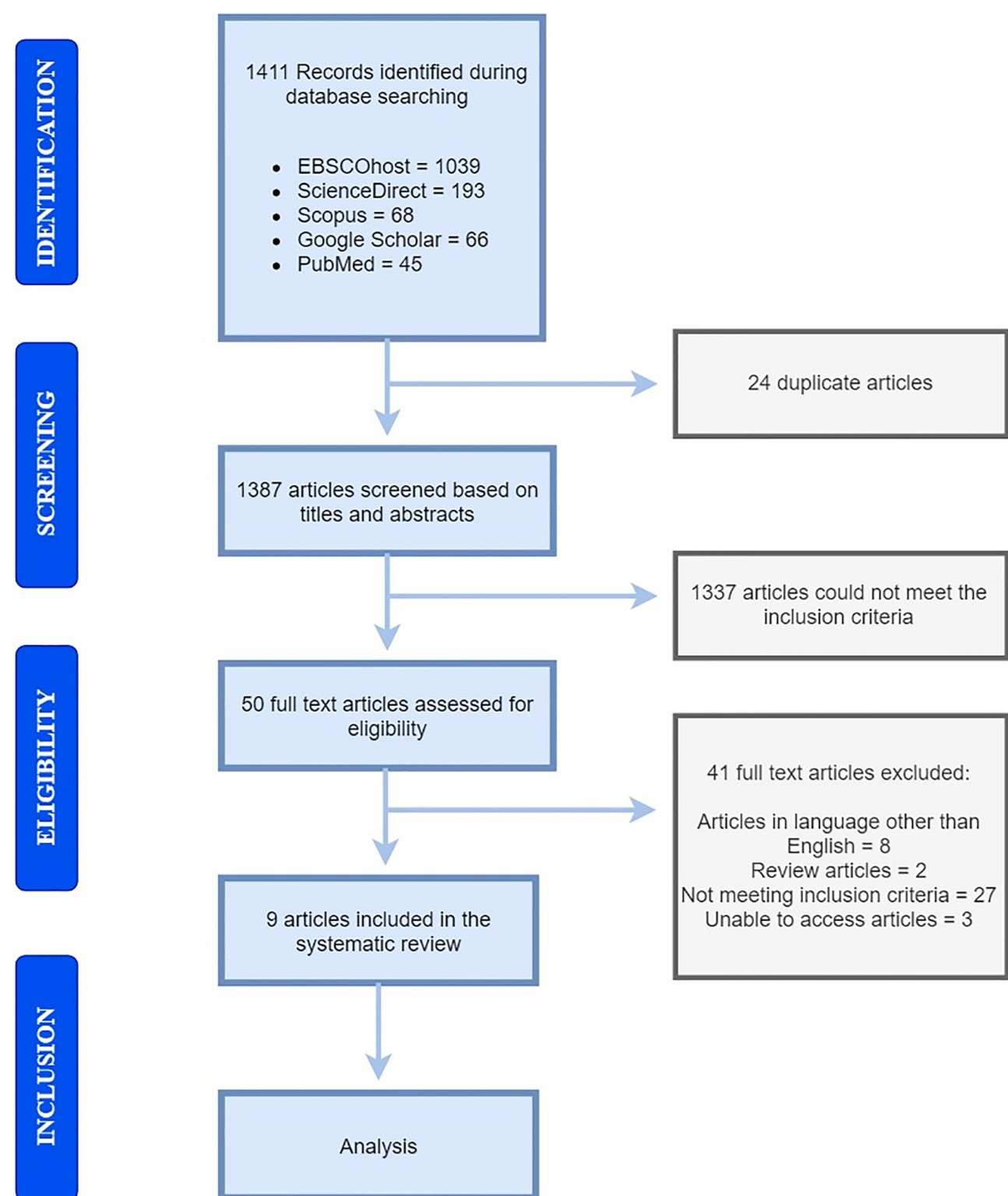

**Fig 1. Schematic diagram showing the assortment of studies and study selection process (2009 PRISMA flow diagram) [8].**

**Table 1. Quality assessment using the Newcastle Ottawa Scale and AHRQ guidelines [9, 10].**

| First Author | Selection | | | | Comparability | Outcome | | Total quality scores | Quality rating according to guidelines[a] |
|---|---|---|---|---|---|---|---|---|---|
| | Representativeness of the sample | Sample size justified/ Not justified | Non-Respondents | Ascertainment of the exposure (Risk factor) | Comparability of different outcome groups | Assessment of the outcome | Statistical test | | |
| D'Agata (2000) [11] | * | Nil | * | Nil | * | ** | * | 6 | Fair |
| MatíasGnass (2014)[12] | * | Nil | * | Nil | * | ** | * | 6 | Fair |
| Luyu Tang (2016)[17] | * | Nil | * | * | * | ** | * | 7 | Good |
| Jae-Uk Song (2017)[13] | * | Nil | * | Nil | * | ** | * | 6 | Fair |
| Geoffrey Taylor (2003)[14] | * | Nil | * | Nil | * | ** | * | 6 | Fair |
| Ping-huai Wang (2016)[15] | * | Nil | * | Nil | * | ** | * | 5 | Fair |
| CokordaAgung (2016)[16] | * | Nil | * | Nil | * | ** | * | 7 | Fair |
| Hussein Yousif (2018)[18] | * | Nil | * | * | * | ** | * | 7 | Good |
| Komal Patel (2015)[19] | * | Nil | * | * | * | ** | Nil | 7 | Good |

[a]NOS rating conversion to AHRQ standards (good, fair, and poor):

For good quality, a study must have 3 or 4 stars, 1 or 2 stars and 2 or 3 stars in 'Selection', 'Comparability' and 'Outcome' domains respectively.

For fair quality, a study must 2 stars, 1 or 2 stars, and 2 or 3 stars in 'Selection', 'Comparability' and 'Outcome' domains respectively.

For poor quality, a study must have 0 or 1 star, 0 star, or 0 or 1 star in 'Selection', 'Comparability' and 'Outcome' domains respectively.

A maximum of one star can be awarded for each subcategory within the Selection and Outcome categories. A maximum of two stars can be awarded for Comparability category.

Two studies were conducted specifically on HD associated blood stream infections (HABSI) [12, 14], while another two studies were on HD associated pneumonia (HDAP) [13, 15]. Erika et al. 2000, reported risk factors related to four nosocomial infections, including, pneumonia, urinary tract infections (UTIs), bloodstream infections, and diarrhoea [11]. Various risk factors associated with all types of nosocomial infections were studied by Tang et al. 2016. [17].

The most frequent pathogens associated with nosocomial infections varied among different studies. Erika et al. 2000, found that candida species (spp) and enterococci were the most common pathogens involved in infections among patients undergoing chronic HD [11]. Another study identified staphylococcus as the main pathogen causing HD-associated nosocomial infections, followed by candida spp [12]. Jae-Uk et al. 2017, found that of the total 53 pathogens responsible for nosocomial infections, the main pathogen involved was *Staphylococcus aureus* (16.1%), followed by *Klebsiella pneumoniae* (10.4%) and *Streptococcus pneumonia* (9.5%) [13]. Similarly, the microbial aetiology of nosocomial infections found by Taylor et al. 2003, were coagulase-negative staphylococci 45%, *S. aureus* 28.1%, enterococcus 8.8%, and aerobic gram negative bacilli 8.6% [14].

Most of the studies included in this review studied one particular nosocomial infection, such as HD-associated bloodstream infection (HABSI), hospital acquired pneumonia (HAP), and certain viral infections [12–16, 18, 19]. Erika et al. 2000, showed that out of 47 episodes of

**Table 2. Study characteristics.**

| Author | Year and study duration | Study design | Respondents | Demographics | Type of infections | Type of microorganismsn (%) | Percentages of nosocomial infections n (%) | Risk factors |
|---|---|---|---|---|---|---|---|---|
| D'Agata et al. [11] | 1995 to 1997 30 months | Retrospective cohort and case control study | Case group -ESRD patients undergoing HD with nosocomial infection Control group—ESRD patients on HD without nosocomial infections | N = 365 Patients for overall study N = 58 patients for Case-control study Mean age = 55 (SD = 8) years | - Pneumonia, - Primary bloodstream infections, - UTIs, - Diarrhoea | Enterococcus spp: 6 (13%), Candida spp: 8 (17%), Enterobacter spp: 6 (13%), Pseudomonas aeruginosa: 5 (10%), Staphylococcus aureus: 3 (6%), and Escherichia coli: 3 (6%) | Total 47 (100%) UTIs 22 (47%), BSI = 13 (28%), Pneumonia 9 (19%), Clostridium difficile diarrhoea 3 (6%) | - **Chronic HD** (RR, 2.4; 95% CI, 1.8 to 3.2; P—0.001). |
| MatíasGnass et al. [12] | 2011 to 2012 24 months | Cross sectional study | Patients on HD | N = 619 Mean (SD) age: 57 (15) years | - HABSIs | Total 14 (100%)–gram positive cocci: 9 (64%) with a predominance of 8 staphylococcal infections Candida spp: 3 (21%). | HABSI 14 (100%) | - **Length of hospital stay** (r = 0.82; 95% CI: 0.79–0.83, P <0.05) - **Number of HD sessions** (r = 0.56; 95% CI: 0.52–0.61, P <0.05) - **HbA1c levels greater than 7%** (OR, 3.62; 95% CI: 1.15–11.4, P <0.05). |
| Tang et al. [17] | 2012 to 2014 36 months | Cross sectional and longitudinal study (Mixed study design) | Patients on HD | N = 890 | All nosocomial infections | Total 98 (100%)—Gram negative bacilli—47 (47.94%), Gram positive bacilli—44 (44.9), and Fungi—7 (7.14%) | Total 110 (100%)–LRTI = 43 (39%), URTI = 23 (20.9%), - catheter-related infection19 (17.27%), - UTIs 13 (11.81%), - and GIT infection. 4 (3.6%) | - **Multiple comorbidities** (OR, 1.66; 95% CI:1.35~2.49) - **Longer duration of HD** (OR, 1.79; 95% CI: 1.35~2.59) - **More catheter sites (two or more)** (OR, 1.12; 95% CI: 1.02~1.85) - **Low Hb concentration** (OR, 0.19; 95% CI: 0.09~0.33) - **Low WBC count** (OR, 0.3; 95% CI: 0.14~0.42) - **Long duration of catheterization** (OR, 1.2; 95% CI: 1.09~1.44) |
| Jae-Uk et al. [13] | 2011 to 2015 60 months | Cross sectional study | Patients on HD | N = 105 Median age—71 years | - HDAP | Total 53 (100%)–S. aureus 17 (16.1%) K. pneumoniae 11 (10.4%), S. pneumoniae 10 (9.5%). | HDAP (100%) | - **PSI score > 147** (OR, 1.023; 95% CI: 1.005–1.041) - **Recent hospitalization** (OR, 2.951; 95% CI:1.022–8.518) |
| Taylor et al. [14] | 1998 to 1999 6 months | Longitudinal or Prospective studies | Group 1: All new patients on HD Group 2: HD with a new vascular access | N = 527 (Cohort study) N = 186 (Case control study) | Blood stream infection (BSI) | Total 96 (100%); coagulase-negative staphylococci 45%, S aureus 28.1%, enterococcus 8.8%, aerobic gram-negative bacilli 8.6%. | HABSI—100% | - **Prior BSI** (OR, 6.56; P = .004), - **Poor patient hygiene** (OR, 3.48; P = .001), - **Contiguous infection** (OR, 4.36; P = .002). |
| Wang et al. [15] | 2005 to 2010 72 months | Cross sectional study | They were enrolled if they fulfilled the criteria for HCAP | N = 530 (48 HD, 482 Non-HD) Mean Age = 68.3 ± 11.3 (HD group), 75.8 ± 12.8 years (Non-HD group) | - HCAP | Total 48 (100%)–β-Streptococcus 2(4.2%), K. pneumoniae 2(4.2%), E. coli 3 (6.3%) Causative pathogens resistant to CAP antibiotics regimen MRSA 5(10.4%), P. aeruginosa 8(16.7%) | HDAP (100%) | - **Wound care** (OR, 4.73; 95% CI: 1.13–19.7), - **Old age (more than 70 years)** (OR, 3.81; 95% CI: 1.07–13.5) - **PSI V** (OR, 3.49, 95% CI: 1.08–12.1). |

*(Continued)*

**Table 2.** (Continued)

| Author | Year and study duration | Study design | Respondents | Demographics | Type of infections | Type of microorganismsn (%) | Percentages of nosocomial infections n (%) | Risk factors |
|---|---|---|---|---|---|---|---|---|
| Agung et al. [16] | 2016 4 months | Cross sectional study | All hospitalized patients who underwent HD. | N = 267 Mean (SD) age = 54.07 (0.80) years | Hepatitis B infection | HBV (100%) | Hepatitis B infection (100%) | - **Duration of HD** (OR, 1.07; 95% CI: 1.03–3.74) - **History of previous transfusion** (OR, 2.49; 95% CI: 1.29–8.18) |
| Hussein et al. [18] | 2015 to 2016 6 months | Cross sectional study | All hospitalized patients who underwent HD. | N = 510 Mean (SD) age = 48 (12) years | Hepatitis B and C infection | HBV and HCV (100%) | Hepatitis B and hepatitis C infection (100%) | **HBsAg positivity:** - **Young age (mean age: 39.2 ± 14.6 years)** ($P < 0.04$), - **History of HD** ($P = 0.005$). **HCV positivity:** - **Old ages (mean age: 55 ± 12 years)** ($P < 0.05$), - **Longer duration of HD** ($P < 0.05$), - **History of surgical and dental procedures** ($P < 0.05$) |
| Patel et al. [19] | 2014 to 2015 15 months | Longitudinal or Prospective studies | Patients who were on HD for a minimum period of 1 month and were likely to be available for follow-up for at least 6 months, were included in the study. | N = 170 | Hepatitis B and C infection | HBV and HCV (100%) | Hepatitis B and Hepatitis C infection (100%) | - **Positive history of blood transfusions** - **Number of blood transfusions** |

HD = Hemodialysis, ESRD = End stage renal disease, UTI = Urinary tract infection, LRTI = Lower respiratory tract infection, URTI = Upper respiratory tract infection, HCAP = Healthcare associated pneumonia, HDAP = Hemodialysis associated pneumonia, CAP = Community acquired pneumonia, GIT = Gastrointestinal tract, BSI = Bloodstream infection, MRSA = Meticillin resistant staphylococcus aureus, HBV = Hepatitis B virus, HCV = Hepatitis C virus, HBsAg = Hepatitis B surface antigen, HABSI = Hemodialysis associated bloodstream infection, WBC = White blood cell, HB = Hemoglobin, N = Sample size, PSI = Pneumonia severity index, SD = Standard deviation, OR = Odd ratio, CI = Confidence interval, RR = Relative risk.

nosocomial infections, nosocomial UTIs were the most common nosocomial infection among HD population, accounting for nearly half of these infections (47%), followed by bloodstream infections (28%), and pneumonia (9%) [11]. Another study reported 110 cases of nosocomial infection occurred, including 66 cases of respiratory tract infections (RTIs), 19 cases of catheter-related infections, 13 cases of urinary tract infections, 4 cases of gastrointestinal infections, and 8 cases of infections in other organ systems [17].

## Outcomes

Some studies reported that longer duration of HD was a risk factor associated with nosocomial infections [16–19], while a positive correlation was determined between HABSI and diabetes, number of HD sessions, and length of hospital stay [12]. Risk for a HABSI increased significantly with HbA1c levels above 7% [12]. There were studies which showed a significant relationship between nosocomial infections and age of patients [15, 16, 18], with HBV infections being more common in relatively younger individuals (mean age: 39.2 ± 14.6 years), while there was a strong relationship between HCV infections and relatively older age group (mean age: 55 ± 12 years) [16, 18]. A positive history and number of blood transfusions were both

significantly associated with HBV or HCV infection [16, 19]. Moreover, patients with multiple co-morbidities, with more catheter sites (two or more), low hemoglobin concentration, low white blood cells count, and longer duration of catheterization were found to be at more risk of developing nosocomial infections [17]. Another study showed that history of bloodstream infections, contiguous infections at surgical site and poor patient hygiene were all independently associated with the occurrence of bloodstream infections [14].

## Discussion

The comprehensive search strategy applied in this review identified nine studies which described the characteristics of patients and risk factors associated with nosocomial infections in ESRD patients receiving HD. Except for one study [11], the results of all the studies were consistent with a positive direction of association between several risk factors and the occurrence of infections in hemodialysis HD patients [12–19]. This review used sensitive search strategy with a broad definition of ESRD, for a comprehensive and inclusive search. The results of this review are consistent with previous narrative reviews that is, an association between ESRD and infections is likely, but still there is a scarcity of the available evidence regarding the risk factors and nosocomial transmission of these infections. Based on the published literature, several risk factors associated with nosocomial infections in HD population were identified. These included factors indicating reduced patient health status such as co-morbidities, poor patient hygiene, prior infections, advanced age, low WBCs count, low hemoglobin, and catheterization. Other important risk factors included longer hospital stay, longer duration on HD, and more HD sessions. The risk factors identified are biologically convincing, which suggests that patients with more health problems that is, those with multiple co-morbidities, have higher in-hospital exposure time, and/or are undergoing longer and more complex procedures are at an increased risk of acquiring nosocomial infections. Major risk factors identified in the current review are depicted in Fig 2.

Longer duration on HD is an important risk factor for nosocomial infections in HD population [16, 18, 19], since patients on chronic HD are at a higher risk of exposure to pathogens than those patients with lesser time on HD. Furthermore, an increase in the duration of dialysis increased the number of venipuncture events and therefore, the risk of related infections [17]. These results are consistent with that of previous studies which showed that the duration of dialysis was longer for patients who were seroconverted than those who remained seronegative for HBV and HCV infections [20–24]. The high incidence of infections in patients on HD for 2 years or more suggests that conditions in the hemodialysis HD unit might have contributed to nosocomial infections [25]. However, the duration on HD is not a modifiable factor and hence, the focus should be on other modifiable factors that can be altered through the implementation of strict infection control guidelines. In the present review, HABSI was significantly associated with risk factors including length of hospital stay and number of HD sessions [12]. Both of these factors directly exposed the patients to the infection agents for prolonged period of time, hence making them more prone to nosocomial infections.

In the present review, patients with diabetes were at higher risk of getting a nosocomial infection than non-diabetes patients [12]. The same was found with increasing HbA1c values above 7% [12]. Better glycemic control has been associated with longer survival in diabetes patients with ESRD undergoing HD, and poor glycemic control increases mortality due to infectious diseases [26–28]. Furthermore, HbA1c $\leq$ 8% and good glycemic control provide reasonable protection against infections caused by hyperglycemia [29]. It has been well documented that diabetes patients on HD have higher risk of morbidity and mortality due to infections compared to non-diabetes patients [29, 30].

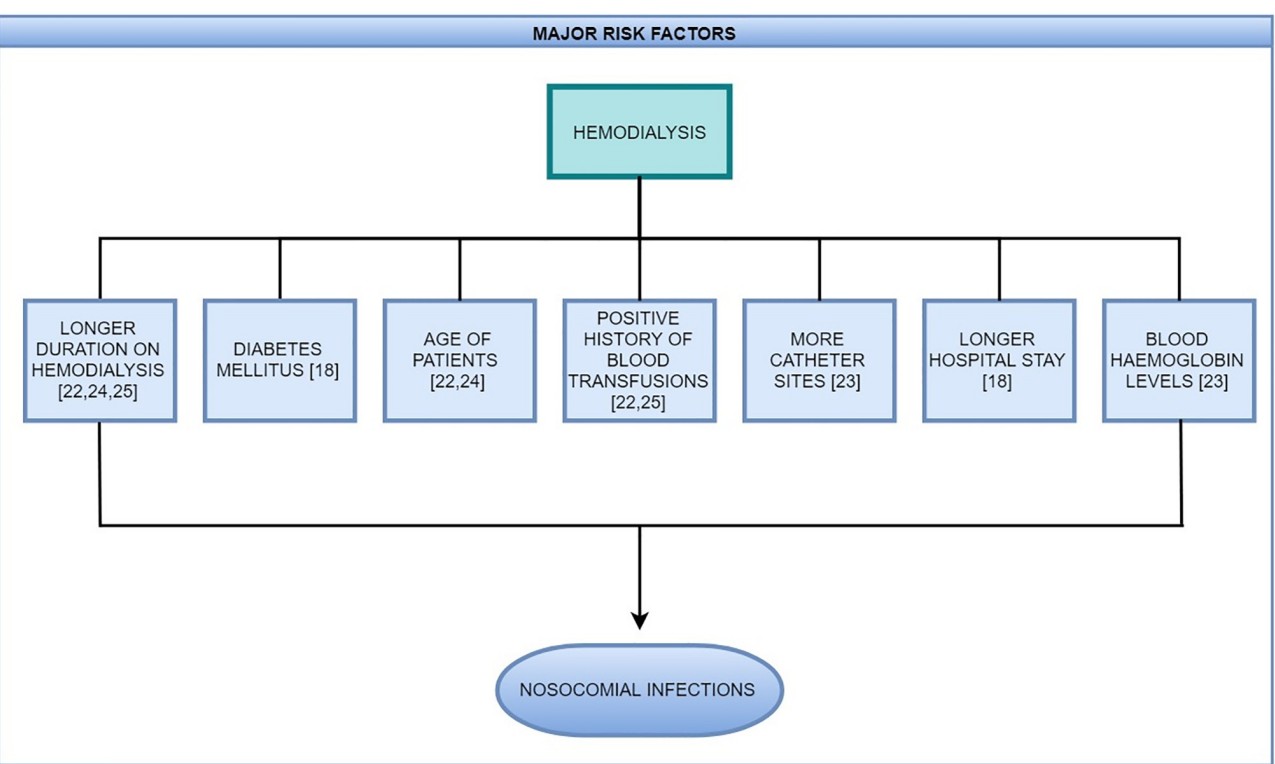

**Fig 2. Major risk factors identified in this review.**

Hyperglycemia caused many adverse effects on immune system mechanisms, both cellular and humoral [31]. Moreover, increased virulence of pathogens, reduced production of interleukins in infection, decreased chemotaxis and phagocytosis, immobilization of neutrophils, glycosuria, urinary and gastrointestinal dysmotility are the main pathogenic mechanisms for infections to occur in the hyperglycemic environment [31]. This implies that good glycemic control reduces the risk of infections in diabetes patients with ESRD and on HD. Even in non-diabetes patient population, stress due to HD often leads to hyperglycemia and hence, this should be monitored and treated. Efficient management of glycemic levels can help to reduce the occurrence of infections.

The results of the present review show that HBV infection is more common in relatively younger patients [16, 18]. This is in agreement with another study done in Libya [32], but the reason for this is unclear. On the other hand, patients infected with HCV were on an average older than non-infected individuals [18]. This observation is congruent with other studies, which showed that the prevalence of HCV infection in HD patients was higher in age group of 40 to 50 years [22, 33]. This may be due to the higher rate of renal diseases in older people than younger people. The aggregation of clinical co-morbidities with advancing age (as on an average, those who are 75 year old have 3 to 4 chronic diseases) [34], management of HD-related issues such as vascular access and infection complications, must be addressed with caution. Therefore, clinicians must incorporate screening and treatment strategies for various HD associated complications in elderly people, including infections, in their routine plan of care.

A positive history and number of blood transfusions are both significantly associated with HBV or HCV infections [16, 19]. Previous studies have shown that the risk of acquiring HBV or HCV infections increases with an increase in the number of blood transfusions [35–37].

Before the use of an effective screening process, blood transfusions were recognized as a major source of nosocomial transmission of HBV and HCV infections. It is still possible that blood donors with HBV or HCV are being overlooked by the current screening procedures and this may need to be readdressed [38]. Therefore, thorough screening of blood donors for HBV and HCV is essential to prevent nosocomial transmission.

Patients with two or more catheter sites or longer catheterization duration are significantly more likely to have nosocomial infections than those with one catheter site and shorter catheterization duration [17, 39, 40]. Catheter related infection (CRIs) in HD patients is one of the major causes of increased morbidity, mortality, and cost of therapy [41–43]. The most effective prevention strategy for these infections is to reduce the use of catheters [44]. Efforts should be made through patient education and vascular access coordinator to reduce the use of catheters by identifying and addressing barriers to permanent vascular access placement and catheter removal [45]. Centers for disease control (CDC) has also recommended other interventions such as hand hygiene, patient and staff education and skin antisepsis to decrease the risk of these infections [46].

In regard to type of vascular access, a study has shown that infection rates were highest for non-tunneled catheters (5.5 per 1,000 hemodialysis sessions), followed by AV grafts (13.51 per 1,000 hemodialysis sessions), tunneled catheters (2.39 per 1,000 hemodialysis sessions), and AV fistula (0.9 per 1,000 hemodialysis sessions) [12]. Another study reported that infection rates per 10,000 procedures were 40.26 for uncuffed CVCs and 45.26 for cuffed CVCs, 7.97 for arteriovenous grafts, and 5.02 for arteriovenous fistula [14].

The present study also showed that low blood Hb concentration was associated with nosocomial infections [17]. Previous observational study which investigated risk factors for bloodstream infections (BSIs) in HD patients concluded that low Hb concentration (approximately 105 g/L) was significantly associated with these infections [47]. One of the possible explanations is the prevalence of malnutrition in dialysis patients, as evidenced by reduced blood Hb and albumin which may lead to dysfunction of the immune system and hence, an increase risk of nosocomial infections [17]. Secondly, since the renal function is compromised, the kidneys decrease or cease the production of erythropoietin which leads to anemia or low Hb in ESRD patients [48]. Thus, the provision of nutritional support for patients on HD is important to help patients improve their immunity function.

Despite the worldwide spread of nosocomial infections, data regarding the risk factors associated with such infections is still lacking. The results of this review suggest a gap and potential benefit of additional preventive options to further reduce the risk of infections in HD population. Moreover, several patient-related and facility-related risk factors have been consistently identified in the studies under this review, which may help to initiate strategies to achieve optimal control measures. Based on the results, widespread adoption of preventive measures and system development should be undertaken to ensure better control of these infections. The comprehensive nature of risk factor consideration and study eligibility criteria are strengths of this review. All observational studies regarding risk factors associated with nosocomial infections among HD population were considered for inclusion in this review. Patient and facility related parameters were obtained from each study, which provided a broad view of risk factors as observed across various clinical settings.

The comprehensive nature of this review also posed some limitations. The studies included in this review varied in the study designs, including retrospective cohort and case control, cross-sectional and longitudinal or prospective study. The heterogeneity nature of the studies included in this review, with various methodologies and risk factors presented challenges in quantitative synthesis of the results. Therefore, only qualitative synthesis was performed as the type of data obtained was not appropriate for meta-analysis. As such, the interpretation of

results was mainly focused on the direction of effect, as opposed to its magnitude. Results of this review summarized the range of studies, and differences in sample size, study designs, time period and quality of studies were not taken into consideration. Although, the variability between different studies hindered a single quantitative estimation for individual risk factors, the review provided support in evidence of the association of factors, such as longer duration on HD, advancing age, positive history of blood transfusion, which were consistently identified as risk factors for nosocomial infections.

Due to a lack of research in the area and despite using broad definition of risk factors and comprehensive search terms, only 9 studies could be identified for inclusion in this review. Therefore, more research is needed to fill the information gap pertaining to nosocomial infections in HD patients. Future research should focus on the risk factors and their control measures to decrease the prevalence of nosocomial infections in HD population. Moreover, more studies are needed before a quantitative estimation of individual risk factors and its impact in this high-risk population can be determined.

## Conclusions

This comprehensive and thorough review of published literature revealed many factors that can contribute towards the occurrence of nosocomial infections in HD population. Controlling and minimizing the effects of these factors will not only improve the patients' health related quality of life (HRQoL) but will also decrease economic burden. Increasing life expectancy and improving HRQoL are the two main health targets in patients with ESRD and can be achieved by minimizing infection complications in this population. Collaborative effort among health care professionals (nephrologists, pharmacists, nurses), caregivers and patients is needed to overcome this health-related issue. In addition to providing a conventional therapy, health care professionals should take special measures to minimize the risk of infections in these patients. The relationship between ESRD and nosocomial infections is intriguing and needs more research to better understand individual risk factors and to develop optimum control measures for them.

## Appendix

### I. List of major exclusions:

1. Incidence and risk factors for bloodstream infections stemming from temporary hemodialysis catheters [49].

2. Hepatitis B and C infection in haemodialysis patients in Libya: prevalence, incidence and risk factors [32].

3. Prevalence of Vancomycin-Resistant Enterococci colonization and its risk factors in chronic hemodialysis patients in Shiraz, Iran [50].

4. Occurrence of infectious diseases in dialysed patients [51]

5. Hepatitis B virus infection in Haemodialysis Centres from Santa Catarina State, Southern Brazil. Predictive risk factors for infection and molecular epidemiology [52].

6. A prospective study of infections in hemodialysis patients: patients hygiene and other risk factors for infection [53].

7. Data Underscore Risk of Nosocomial Infections in Chronic HD Patients [54].

8. Risk Factors for Infection-Related Hospitalization in In-Center Hemodialysis [55]

9. Risk factors for morbidity and mortality of bloodstream infection in patients undergoing hemodialysis: a nested case–control study [56].

10. Surveillance of chronic haemodialysis-associated infections in southern Israel [57].

11. Prevalence of Hepatitis c virus (HCV) infection and related risk factors among Iranian patients on hemodialysis [58].

12. A prospective study of infections in hemodialysis patients: Patient hygiene and other risk factors for infection [59].

13. Risk factors for catheter-related infections in patients on hemodialysis [60].

14. Viral hepatitis C and B among dialysis patients at the Rabat University Hospital: prevalence and risk factors [61].

15. Hemodialysis catheter-related infection: rates, risk factors and pathogens [42].

16. Incidence and risk factors of bloodstream catheter-related infections in hemodialysis patients [62].

17. Central venous catheter for hemodialysis: incidence of infection and risk factors [63].

18. Risk factors for candidemia with non-albicans Candida spp. in intensive care unit patients with end-stage renal disease on chronic hemodialysis [64].

19. Clinical epidemiology of pneumonia in hemodialysis patients: the USRDS waves 1, 3, and 4 study [65].

20. Risk Factors of HCV Seroconversion in Hemodialysis Patients in Tabriz, Iran [66].

21. Epidemiology of hemodialysis vascular access infections from longitudinal infection surveillance data: Predicting the impact of NKF-DOQI clinical practice guidelines for vascular access [67].

22. Prevalence and risk factors of hepatitis C and B virus infections in hemodialysis patients and their spouses: A multicenter study in Beijing, China [68].

23. Hepatitis C virus infection in hemodialysis or continuous ambulatory peritoneal dialysis patients: Results of comparative analysis [69].

24. Vancomycin-resistant enterococci colonization in patients at seven hemodialysis centers [70].

25. Detection of hepatitis C virus in patients with terminal renal disease undergoing dialysis in southern Brazil: prevalence, risk factors, genotypes, and viral load dynamics in hemodialysis patients [71].

26. Risk factors for nosocomial infections in hemodialysis patients and nursing interventions [72].

27. Analysis and countermeasures of risk factors of nosocomial infection in hemodialysis patients [73].

28. Occurrence rates, risk factors and direct economic losses of healthcare-as-sociated infection in hemodialysis patients in a tertiary first-class hospital [74].

29. Risk Factors and Prevention of Nosocomial Infection Inpatients with Hemodialysis [75].

30. Infection Episode and Related Risk Factors in Continuous Hemodialysis Patients: A Survey [76].

31. Risk Factors of Hospital Infection in Hemodialysis Department: Prevention and Control [77].

32. Clinical features of nosocomial infections in chronic renal failure patients who underwent hemodialysis and related risk factors [78].

33. Risk Factors of Hospital Infection in Hemodialysis Department: Prevention and Control [79].

34. Department of Nephrology The First Affiliated Hospital of Guangdong College of Pharamcy, Guangzhou 510080, China; Analysis of Pulmonary Infection and Related risk Factors in Hemodialysis Patients [80]

35. Distribution and risk factors of infection in patients with maintenance hemodialysis [81].

36. Clinical characteristics and risk factors of nosocomial infection in patients with chronic renal failure Hemodialysis [82].

37. Infection control in hemodialysis units: a quick access to essential elements [83].

38. Multidisciplinary health team in control of risk factors for colonization and infection caused by MRSA in hemodialysis [84].

39. Vascular access infection among hemodialysis patients in Northern Jordan: incidence and risk factors [85].

40. Molecular epidemiology of a hepatitis C virus outbreak in a hemodialysis unit in Italy [86].

41. Prevalence of Vancomycin-Resistant Enterococci Among Children with End-Stage Renal Failure [87].ss

## Supporting information

**S1 Data. Data extraction sheet.**
(XLSX)

## Author Contributions

**Data curation:** Saad Hanif Abbasi.

**Methodology:** Saad Hanif Abbasi.

**Project administration:** Siew Siang Chua.

**Supervision:** Raja Ahsan Aftab, Siew Siang Chua.

**Writing – original draft:** Saad Hanif Abbasi.

**Writing – review & editing:** Raja Ahsan Aftab, Siew Siang Chua.

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
