## [Decision Letter · Decision Letter 0]

29 Jan 2020

PONE-D-19-32738

Risk factors associated with nosocomial infections among end stage renal disease patients undergoing hemodialysis: A systematic review

PLOS ONE

Dear Dr. AFTAB,

Thank you for submitting your manuscript to PLOS ONE. After careful consideration, we feel that it has merit but does not fully meet PLOS ONE’s publication criteria as it currently stands. Therefore, we invite you to submit a revised version of the manuscript that addresses the points raised during the review process.

We would appreciate receiving your revised manuscript by Mar 14 2020 11:59PM. To enhance the reproducibility of your results, we recommend that if applicable you deposit your laboratory protocols in protocols.io, where a protocol can be assigned its own identifier (DOI) such that it can be cited independently in the future. For instructions see: http://journals.plos.org/plosone/s/submission-guidelines#loc-laboratory-protocols

We look forward to receiving your revised manuscript.

Kind regards,

Laura Pasin

Academic Editor

PLOS ONE

Journal Requirements:

3. We noted that your search was done approximately 2 years ago. We ask that you please provide an update to your search to capture any manuscripts that may have been published since.

Reviewers' comments:

Reviewer's Responses to Questions

**Comments to the Author**

1. Is the manuscript technically sound, and do the data support the conclusions?

Reviewer #1: No

Reviewer #2: Yes

Reviewer #3: Yes

Reviewer #4: Yes

Reviewer #5: Yes

2. Has the statistical analysis been performed appropriately and rigorously? 

Reviewer #1: I Don't Know

Reviewer #2: N/A

Reviewer #3: No

Reviewer #4: Yes

Reviewer #5: Yes

3. Have the authors made all data underlying the findings in their manuscript fully available?

Reviewer #1: Yes

Reviewer #2: Yes

Reviewer #3: Yes

Reviewer #4: Yes

Reviewer #5: Yes

4. Is the manuscript presented in an intelligible fashion and written in standard English?

Reviewer #1: Yes

Reviewer #2: Yes

Reviewer #3: Yes

Reviewer #4: Yes

Reviewer #5: Yes

5. Review Comments to the Author

Reviewer #1: Saad Hanif Habbasi has considered 9 studies in order to evaluate the risk factor associated with nosocomial infections among the ESRD ( End-Stage Renal Disease) hemodialyzed patients.

The argument is crucial and has a big impact in the national health services and in the daily life of patients.

However there are some considerations.

Firstly the author should specify the type and the site of vascular access for hemodialysis.

The 2006 KDOQI vascular access guidelines considers AV fistula the first option followed by prothesic grafts if AV fistula is not possible. So the long term vascular catheter should be placed as last choice . Sometimes it could be necessary to occur to a vascular catheter temporarily if fistula is not matured and it should be indicated for how long.

It is well known that vascular catheter for dialysis has a higher risk of bloodstream infections in comparison to AV fistula. In addition patients who are cvc dependent for dialysis are more sick than patients with AV fistula

Secondly the nine studies are conducted in different period of time and this makes results difficult to be interpreted. Recent data show a reduction of CLABSIs (central line–associated bloodstream infection) since 2001 as a result of prevention programs in many hospitals around the world. In additon in the 2016 the CDC ( Centers for Disease Control and Prevention) estimates that CLABSIs decreased by 50% between 2008-2014 .

Finally the risk of bacteremia and infections depends on the intrinsic defence mechanism and therapies of patients. Most of the studies included elderly people who are more exposed to risk of hospitalisation and nosocomial infections because of the accumulation of comorbid disease with age and poor adherence to medical treatment.

Reviewer #2: In this manuscript, dr. Aftab and colleagues performed a systematic literature review aimed at identifying risk factors for noscomial infections in patients with end-stage renal failure undergoing hemodyalisis.

The topic is relevant, given the expected increase in the number of patients who will require intermittent hemodyalisis in the next future, and costs associated with noscomial infections. The manuscript is overall well conducted from a methodological point of view, with multiple databases searched, description of search strategy, independent screening, and risk of bias assessment.

The major flaw of this study is lack of a statistical analysis of the results, although I acknowlegde that the type of study make it difficult to perform a quantitative analysis.

I have a few comments for the Authors which I hope will help them to improve their manuscript:

1. The introduction is too long. I suggest to shorten it, and to focus on why the Authors decided to undertake this study

2. Given that no quantitative analysis was performed, the paragraph can be deleted. A sentence a the end of the Methods section is sufficient

3. Please provide a list of the 41 major exclusions, together with references, in a Supplementary Appendix

4. Please rewrite the Discussion, underlying the relationship of present study to previous literature, the significance of study findings, what present study add to current knowledge, and future directions. The detailed discussion of each risk factor is not necessary

Minor comment

1. in the paragraph "Characteristics of the Selected Studies" on lines 11-13 the Authors write "in some studies, there were participants aged 12 years and above". As per exclusion criteria, all studies with participants aged < 12 should have been excluded.

Reviewer #3: I am grateful for the opportunity to review the manuscript entitled "Risk factors associated with nosocomial infections among end stage renal disease patients undergoing hemodialysis: A systematic review ".

I have some comments for the author. I hope that my comments may contribute improving your papers.

Manuscript is well written, really clear and sections length is adequate. Moreover,reference are uptodate.

Nonetheless, there are two main issues.

1)There are few studies included. Nine studies are a limited number for a systematic review. Moreover, studies included are different in study design, population, inclusion/exclusion criteria and ESRD definition.

2)Lack of meta-analysis is probably the major issue of this paper. Without a proper analysis I believe most of your results and discussion remain unproven. I would suggest author to try to synthetize results whenever possible for key outcomes.

Introduction:You begin your manuscrit with the second reference, please number the reference accordingly to first appearance.

Minor issues:

Data extraction: I think it would be useful to include the data extraction form as supplementary material

Risk of Bias/Quality: Explain your strategy to overcome disagreements between assessors

Data synthesis: "A systematic review was performed to make sure that all data synthesis done was sourced from the maximum possible, and complete collection of relevant literature" I believe this sentence may expanded in order to clarify your efforts. (Examples: reference check, contacting the authors, contacting fields experts)

Results: "In some studies, there were participants aged above 12 years" The sentence is not clear. Inclusion criteria is >12 years, so all participants have to be aged > 12 years old. Please explain.

Reviewer #4: There’s no doubt we should consider this systematical review as an original research: ESRD people are increasing especially in front of life expectancy extension. Health care world may consider main infection risk factors in order to decrease morbidity, mortality and cost effectiveness. Despite we can trace principle elements implicated in ESRD nosocomial infection we don’t know exactly what we have to be afraid of and if we are able to prevent them.

As the author show, a lot of literature meet key words research, but as a matter of facts only few articles really describe patient’s characteristics and nosocomial infections risk factors. If they do usually concentrate themselves on a particular risk factor or subtype of population.

We can declare that results have not been published elsewhere and they have been obtained by proper/ analyses. We found in Cochrane library only a single review focused on “Interventions for preventing infectious complications in hemodialysis patients with central venous catheters” but it doesn’t consider patients characteristics and risk factors.

In this work Authors described in detail the protocol of their systematical review. They declared to follow PRISMA guidelines for the design study and they subscribe the protocol to PROSPERO specifying the registration number.

As a matter of fact the study is conformed to PRISMA statement. Every part is described in sufficient detail and flow chart is clear.

Some data about sources choose has been delete. On the other hand quality of the main studies has been evaluated and described following Otawa criteria. Data extraction form has been good described, considering items author’s name, year and duration of the study, study design, respondents, demographics of the patients (sample size, gender, and age), type of nosocomial infections studied, type of pathogens involved, and risk factors associated with nosocomial infections.

The article adheres to appropriate reporting guidelines and community standards for data availability.

We suggest minor revisions:

1) Please, report the systematic review registration number on PROSPERO in the abstract

2) Risk of bias assessment has been correctly evaluated with Newcastle Ottawa scale (NOS) for observational studies considering three categories: “Selection,” “Comparability,” and “Outcome. Studies have been classified as good, fair and poor quality. It would be useful to describe single limitations of the studies analyzed. In particular 2 have retrospective design and declare this aspect on their discussion.

3) An accurate and comprehensive systematic review has been performed and complete collection of relevant literature. Only qualitative analysis has been undertaken and not statistical analysis has been performed. Authors declare the limitations of their work for heterogeneity nature of the studies considered (retrospective cohort and case control, cross-sectional and longitudinal, prospective). However, it would be interesting to approach a univariate/multivariate analysis of similar studies to evaluate risk ratio or ODDs ratio for hemoglobin level/previous blood transfusion or HD duration in HBV/HCV infection (cross sectional studies). Moreove, HDAP rate, microorganism type, and infection risk in Chronic HD it would be obtained.

4) Methods are exhaustive described, but it would be useful to specify if references authors had been called looking for other works.

5) The manuscript meets the criteria of linguistic clarity and correctness required except for a wrong transcription: table 2, reference 17 (“D’Agata EMC et al, instead of: Erika et al”)

Reviewer #5: Please see attached word document for specific review comments on this manuscript. Thank you for the opportunity to review. I recommend a 'Minor Revision'.

Many Thanks,

6. PLOS authors have the option to publish the peer review history of their article (what does this mean?). If published, this will include your full peer review and any attached files.

Reviewer #1: No

Reviewer #2: Yes: Alessandro Belletti

Reviewer #3: No

Reviewer #4: No

Reviewer #5: Yes: Pasquale Nardelli

---

## [Author Response · Author response to Decision Letter 0]

7 Apr 2020

S. N. Comments Corrections

Overall comments:

Thank you for your comments. Kindly accept my sincere gratitude for your valuable input. I have made changes in the revised manuscript as advised by you. 

Kind regards,

Specific comments:

Reviewer 1

1. The author should specify the type and the site of vascular access for hemodialysis.

The 2006 KDOQI vascular access guidelines considers AV fistula the first option followed by prothesic grafts if AV fistula is not possible. So, the long-term vascular catheter should be placed as last choice. Sometimes it could be necessary to occur to a vascular catheter temporarily if fistula is not matured and it should be indicated for how long.

It is well known that vascular catheter for dialysis has a higher risk of bloodstream infections in comparison to AV fistula. In addition, patients who are CVC dependent for dialysis are more sick than patients with AV fistula Agreed. Although it is well documented that CVC is associated with increased number of infections among hemodialysis patients, however the type and site of vascular access were not found by any study in this review as risk factors, hence they were not discussed. As this is a systematic review, the data were extracted from 9 studies that met our inclusion criteria and only those risk factors that were found significantly associated (p-value = <0.05) were discussed. 

However, a paragraph is added on type of accesses as suggested. Refer to Page no. 26 and line no. 15.

2. The nine studies are conducted in different period of time and this makes results difficult to be interpreted. Recent data show a reduction of CLABSIs (central line–associated bloodstream infection) since 2001 as a result of prevention programs in many hospitals around the world. In addition, in the 2016 the CDC (Centers for Disease Control and Prevention) estimates that CLABSIs decreased by 50% between 2008-2014. 

Agreed. The heterogeneity among studies included in this review was presented as it’s limitation and was discussed under ‘discussion’ section (Page No. 27 and line No. 23). Due to limited number of studies on nosocomial infections in HD patients, the difference in aspects, such as study designs, methodologies, time period were not taken into consideration.

Secondly, this review considered all the nosocomial infections including CRBSIs. Although, there might be reduction a in CLABSIs due to prevention programs, but still HD population is one of the most high-risk populations for CRBSIs, as indicated by many studies. Therefore, the risk factors for these and other nosocomial infections were elucidated in this review to help in the development of prevention guidelines against these factors. Moreover, CLABSIs is a broad terminology and not used specifically for access used in HD patients.

3. The risk of bacteremia and infections depends on the intrinsic defense mechanism and therapies of patients. Most of the studies included elderly people who are more exposed to risk of hospitalization and nosocomial infections because of the accumulation of comorbid disease with age and poor adherence to medical treatment. According to the inclusion criteria for this review, the studies done on patients of > 12 years of age were included. The studies under this review varied based on age of patients and mean ages ranged from 48 to 71. Age was identified as one of the risk factors in this review and discussed in ‘discussion section’ (Page No. 25 and Line no. 10).

Although, it is well known that with advancing age, there are more chances of acquiring an infection due to various reasons, few studies in this review reported that nosocomial infections were more common in relatively younger age group (Page No. 25 and line No. 10).

In addition, the impact of diabetes as a comorbidity was also discussed separately (Page No. 24 and line No. 15).

Reviewer 2

1. The introduction is too long. I suggest to shorten it, and to focus on why the Authors decided to undertake this study. Agreed. The Introduction has been shortened as recommended.

2. Given that no quantitative analysis was performed, the paragraph can be deleted. A sentence at the end of the Methods section is sufficient Agreed and changes made as suggested. A sentence has been added at the end of the paragraph to indicate this: Only qualitative analysis was undertaken (Page no.7 and line no. 22)

3. Please provide a list of the 41 major exclusions, together with references, in a Supplementary Appendix Agreed. 41 major exclusions are stated in the Supplementary Appendix with references (Page no. 36).

4. Please rewrite the Discussion, underlying the relationship of present study to previous literature, the significance of study findings, what present study add to current knowledge, and future directions. The detailed discussion of each risk factor is not necessary. Agreed and the Discussion section has been revised to include reviewer’s suggestions. 

5. In the paragraph "Characteristics of the Selected Studies" on lines 11-13 the Authors write "in some studies, there were participants aged 12 years and above". As per exclusion criteria, all studies with participants aged < 12 should have been excluded. Agreed. That sentence showed a repetition, and therefore was removed.

Reviewer 3

1. There are few studies included. Nine studies are a limited number for a systematic review. Moreover, studies included are different in study design, population, inclusion/exclusion criteria and ESRD definition. Agreed. The heterogeneity among the studies included in this review was presented as it’s limitation and was discussed under ‘discussion’ section (Page No. 27 and line No. 23). Due to lack of research and limited number studies on nosocomial infections in HD patients, the difference in aspects, such as study designs, methodologies, time period were not taken into consideration.

However, this review included studies on hemodialysis patients so ESRD definition and target population did not affect heterogeneity among the studies.

2. Lack of meta-analysis is probably the major issue of this paper. Without a proper analysis I believe most of your results and discussion remain unproven. I would suggest author to try to synthetize results whenever possible for key outcomes. Agreed. However, only descriptive analysis was performed as the type of data obtained was not appropriate for meta-analysis. The studies in this review had used different statistical methods. Some studies reported risk factors using odd ratios (OR) and confidence interval (CI), while others used correlation coefficients (r), relative risk (RR) and simple p-values to portray their results, as shown in Table 2 in a manuscript. Hence, no risk factors could be grouped together for quantitative synthesis. 

3. Introduction: You begin your manuscript with the second reference, please number the reference accordingly to first appearance.

 Agreed and changes have been made in the revised manuscript.

4. Data extraction: I think it would be useful to include the data extraction form as supplementary material. Agreed and data extraction sheet has been added as a supplementary material.

5. Risk of Bias/Quality: Explain your strategy to overcome disagreements between assessors. Agreed and this has been added in the revised manuscript (Page no. 7 and line no. 16).

6. Data synthesis: "A systematic review was performed to make sure that all data synthesis done was sourced from the maximum possible, and complete collection of relevant literature" I believe this sentence may expanded in order to clarify your efforts. (Examples: reference check, contacting the authors, contacting fields experts). Agreed and the information suggested has been added in the revised manuscript (Page no. 7 and line no.20).

Reviewer 4

1. Please, report the systematic review registration number on PROSPERO in the abstract. Agreed and the information suggested has been added in the revised manuscript (Page no. 2 and line no. 17).

2. Risk of bias assessment has been correctly evaluated with Newcastle Ottawa scale (NOS) for observational studies considering three categories: “Selection,” “Comparability,” and “Outcome. Studies have been classified as good, fair and poor quality. It would be useful to describe single limitations of the studies analyzed. In particular 2 have retrospective design and declare this aspect on their discussion. Agreed and the information suggested has been added in the revised manuscript. Refer to page no. 27 and line no. 21 for limitations of the study.

3. An accurate and comprehensive systematic review has been performed and complete collection of relevant literature. Only qualitative analysis has been undertaken and not statistical analysis has been performed. Authors declare the limitations of their work for heterogeneity nature of the studies considered (retrospective cohort and case control, cross-sectional and longitudinal, prospective). However, it would be interesting to approach a univariate/multivariate analysis of similar studies to evaluate risk ratio or ODDs ratio for hemoglobin level/previous blood transfusion or HD duration in HBV/HCV infection (cross sectional studies). Moreover, HDAP rate, microorganism type, and infection risk in Chronic HD it would be obtained. Agreed. However, only descriptive analysis was performed due to the type of data being unsuitable for meta-analysis. The studies in this review had used different statistical methods as some studies reported risk factors using odd ratios (OR) and confidence interval (CI), while others used correlation coefficients (r), relative risk (RR) and simple p-values to portray their results as shown in Table 2 in a manuscript. Hence, no risk factors could be grouped together for quantitative synthesis.

4. Methods are exhaustive described, but it would be useful to specify if references authors had been called looking for other works. Agreed and the information suggested have been added in the revised manuscript (Page no. 7 and line no. 20). 

5. The manuscript meets the criteria of linguistic clarity and correctness required except for a wrong transcription: table 2, reference 17 (“D’Agata EMC et al, instead of: Erika et al”)

 Agreed and changes made in the revised manuscript.

Reviewer 5

1. Ref. 1 is not called out in order the text. Please fix. Agreed and changes made in revised manuscript.

2. P. 10 L 11-13: I cannot understand. Please reword Agreed and changes made in revised manuscript. 

3. P. 30 L 7-10: consider rewording Agreed and changes made in revised manuscript.

---

## [Decision Letter · Decision Letter 1]

27 May 2020

Risk factors associated with nosocomial infections among end stage renal disease patients undergoing hemodialysis: A systematic review

PONE-D-19-32738R1

Dear Dr. Raja Ahsan Aftab,

We are pleased to inform you that your manuscript has been judged scientifically suitable for publication and will be formally accepted for publication once it complies with all outstanding technical requirements.

With kind regards,

Laura Pasin

Academic Editor

PLOS ONE

Additional Editor Comments (optional):

Reviewers' comments:

Reviewer's Responses to Questions

**Comments to the Author**

1. If the authors have adequately addressed your comments raised in a previous round of review and you feel that this manuscript is now acceptable for publication, you may indicate that here to bypass the “Comments to the Author” section, enter your conflict of interest statement in the “Confidential to Editor” section, and submit your "Accept" recommendation.

Reviewer #2: All comments have been addressed

Reviewer #4: All comments have been addressed

Reviewer #5: All comments have been addressed

2. Is the manuscript technically sound, and do the data support the conclusions?

Reviewer #2: Yes

Reviewer #4: Yes

Reviewer #5: Yes

3. Has the statistical analysis been performed appropriately and rigorously? 

Reviewer #2: Yes

Reviewer #4: N/A

Reviewer #5: Yes

4. Have the authors made all data underlying the findings in their manuscript fully available?

Reviewer #2: Yes

Reviewer #4: Yes

Reviewer #5: Yes

5. Is the manuscript presented in an intelligible fashion and written in standard English?

Reviewer #2: Yes

Reviewer #4: Yes

Reviewer #5: Yes

6. Review Comments to the Author

Reviewer #2: In this manuscript, dr. Aftab and colleagues present a revised version of their manuscript

I believe that all of my comments have been adequately addressed

Reviewer #4: (No Response)

Reviewer #5: I thank the authors for their efforts. I believe the manuscript quality was improved in the review process.

No further comments.

7. PLOS authors have the option to publish the peer review history of their article (what does this mean?). If published, this will include your full peer review and any attached files.

Reviewer #2: Yes: Alessandro Belletti

Reviewer #4: No

Reviewer #5: Yes: Pasquale Nardelli, MD

---

## [Editor Report · Acceptance letter]

9 Jun 2020

PONE-D-19-32738R1 

Risk factors associated with nosocomial infections among end stage renal disease patients undergoing hemodialysis: A systematic review 

Dear Dr. Aftab:

I'm pleased to inform you that your manuscript has been deemed suitable for publication in PLOS ONE. Congratulations! Your manuscript is now with our production department. 

Kind regards, 

on behalf of

Dr. Laura Pasin 

Academic Editor

PLOS ONE